# LoRA-FA: Memory-efficient Low-rank Adaptation for Large Language Models Fine-tuning

## Abstract

The low-rank adaptation (LoRA) method can largely reduce the amount of trainable parameters for fine-tuning large language models (LLMs) and it becomes a very common technique in fine-tuning LLMs. However, during fine-tuning, it still requires very expensive activation memory to update low-rank weights. Though there exist studies trying to reduce the storage of activations, they either would sacrifice model performance or take much longer time for fine-tuning models. To this end, we propose a memory-efficient fine-tuning method, named LoRA-FA, that significantly reduces the activation memory without performance degradation and extra computational costs. Specifically, LoRA-FA freezes the projection-down weight of $A$ and updates the projection-up weight of $B$ in each LoRA layer. It ensures the change of model weight reside in a low-rank space as like LoRA to preserve the fine-tuning performance while eliminating the requirement to store full-rank input activations so as to reduce the overall memory consumption. We conduct extensive experiments across multiple model types (RoBERTa, T5, LLaMA) and model scales. Our results show that LoRA-FA always preserves the fine-tuning accuracy across different tasks and it reduces the overall memory costs by up to $4\times$ and $1.4\times$ compared to full-parameter fine-tuning and LoRA, respectively. Furthermore, LoRA-FA is also compatible with other advanced memory optimization methods like FlashAttention, QLoRA, and ZeRO.

## 1 Introduction

Large language models (LLMs) have become a cornerstone of natural language processing (Brown et al., 2020; Touvron et al., 2023a; OpenAI, 2023; Anil et al., 2023), and fine-tuning pre-trained LLMs has been shown very effective to improve their performance in various downstream tasks (Liu et al., 2019; Wei et al., 2021) and to enable them to align with human intents (Ouyang et al., 2022; Bai et al., 2022). However, fine-tuning LLMs with full parameter is prohibitively expensive, for example, fine-tuning a LLaMA-65B (Touvron et al., 2023a) model with AdamW (Loshchilov & Hutter, 2017) requires more than 1TB of GPU memory to store model parameters, gradients, and optimizer states (Rajbhandari et al., 2020).

To reduce the memory of full-parameter fine-tuning, parameter-efficient fine-tuning (PEFT) methods are proposed to update only a small fraction of parameters, such as adapter weights (Houlsby et al., 2019; Hu et al., 2022) and prompt weights (Li & Liang, 2021; Lester et al., 2021). Among these methods, LoRA (Hu et al., 2022) has shown to achieve comparable performance to full-parameter fine-tuning (Full-FT), and it has been widely used in many applications (Dettmers et al., 2023).

Specifically, LoRA adds a parallel low-rank adapter besides the weight of a linear layer, as shown in Figure 1(b), where $W$ is the pre-trained weight, $A$ and $B$ are low-rank weights. Because LoRA freezes $W$ and only updates smaller matrices $A$ and $B$, its memory overhead for trainable parameter and corresponding gradient and optimizer states can be largely reduced, compared to Full-FT as shown in Figure 1(a), which can be regarded as updating $W$ and freezing $A$ and $B$. Furthermore, LoRA introduces no additional inference latency by merging the value of $AB$ into $W$.

*However, LoRA has a critical limitation that requires expensive activation memory consumption in LoRA layers*. This is because the input of LoRA layer, $X$ (i.e., the activation output of its previous layer) should be stored during the feed-forward pass as it is required to construct the gradient of $A$ during back-propagation. It means that LoRA and Full-FT have the same activation memory cost,

which almost dominates the overall memory consumption during fine-tuning. For example, fine-tuning a LLaMA-65B with an input sequence length of 2048 and a batch size of 4 requires more than 50GB of activation memory (in 16-bit format) in all LoRA layers. Though there exist methods like selecting only a part linear layers as LoRA adapters Hu et al. (2022) and/or using activation recomputation Chen et al. (2016) trying to reduce the activation memory, they are impractical as they either may sacrifice fine-tuning performance (Dettmers et al., 2023) or take much longer time Chen et al. (2016).

To this end, in this work, we propose a novel memory-efficient adapter, LoRA-FA (LoRA with freezing the $A$ matrix), that significantly reduces the activation memory footprint of LoRA without affecting fine-tuning performance. Specifically, LoRA-FA freezes both the pre-trained weight $W$ and the projection-down weight $A$, and only updates projection-up weight $B$, as shown in Figure 1(c). By doing so, LoRA-FA only needs to compute the gradient of $B$, which requires storing a much smaller input of $AX$ during the feed-forward pass, thus eliminate the memory consumption for storing $X$ in LoRA. Assume that $X \in \mathbb{R}^{b \times d}$, $W \in \mathbb{R}^{d \times d}$, $A \in \mathbb{R}^{d \times r}$, and $B \in \mathbb{R}^{r \times d}$, the projection-down weight of $A$ has mapped a $d$-dimensional input of $X$ into an $r$-dimensional input of $XA \in \mathbb{R}^{b \times r}$. As we have $r \ll d$, the activation memory requirement for LoRA-FA can be significantly reduced. For example, LoRA-FA (with a rank size of $r = 4$) reduces the activation memory in a linear layer of LLaMA-65B (with a hidden dimension of $d = 8192$) by 2048 times compared to full-parameter fine-tuning. At the same time, LoRA-FA reduces the amount of trainable parameters from $d^2$ to $dr$ by 2048 times. More details about memory complexity of LoRA-FA are provided in Table 1.

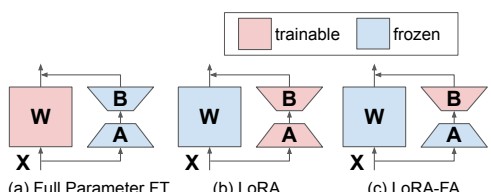

Figure 1: The illustration of (a) full-parameter fine-tuning (FT), (b) LoRA, and (c) LoRA-FA.

We conduct extensive experiments across many model types and scales. We fine-tune RoBERTa (Liu et al., 2019) on natural language understanding tasks, T5 (Raffel et al., 2020) on machine translation tasks, and LLaMA (Touvron et al., 2023a) on MMLU (Hendrycks et al., 2021) benchmarks. Experimental results show that LoRA-FA achieves very close model accuracy across many tasks compared to full-parameter fine-tuning and LoRA. In terms of memory efficiency, our LoRA-FA reduces the overall memory cost by up to $4\times$ and $1.4\times$, compared to full-parameter fine-tuning and LoRA, respectively. For example, compared to full-parameter fine-tuning, LoRA-FA reduces the memory footprint from 112GB to 27.5GB for fine-tuning a LLaMA-7B model.

In addition to the memory efficiency of LoRA-FA in reducing both the amount of trainable parameters and activations, it has several key advantages: 1) it does not increase neither the computational overhead during the fine-tuning stage nor latency during the inference stage, 2) it achieves similar model performance across many models and tasks compared to full-parameter fine-tuning, and 3) it can be combined with advanced memory optimization techniques to further reduce memory footprint in LLMs fine-tuning.

## 2 BACKGROUND AND MOTIVATIONS

### 2.1 LARGE LANGUAGE MODELS

We focus on transformer-based large language models (LLMs). The transformer model was first proposed in (Vaswani et al., 2017) for machine translation task. Later, different transformer models have been used in language modelling (i.e., pre-training), and the pre-trained models are adapted to many downstream tasks (Kenton & Toutanova, 2019; Raffel et al., 2020; Brown et al., 2020).

Take decoder-only GPT (Brown et al., 2020) model as an example, it consists of $L$ stacked transformer blocks, and each block has two sub-modules: multi-head attention (MHA) and feed-forward network (FFN). MHA has three linear layers and they transform the input into query $Q$, key $K$, and value $V$, which are fed into the self-attention for interaction. The output of self-attention is then sent to another linear layer. FFN consists of two linear layers and a GeLU activation function be-

Table 1: Memory complexity comparison among full-parameter fine-tuning (FT), LoRA and LoRA-FA for a single linear layer with mixed-precision training. # TPs is the number of trainable parameters. $d$, $r$, $b$, $s$ are hidden dimension, LoRA rank, batch size, and sequence length, respectively. We calculate the weight, gradient, optimizer, activation in the unit of byte.

| Method | # TPs | Weight | Gradient | Optimizer | Activation |
|--------|-------|--------|----------|-----------|------------|
| Full-FT | $d^2$ | $2d^2$ | $2d^2$ | $12d^2$ | $2bsd$ |
| LoRA | $2dr$ | $2(d^2 + 2dr)$ | $4dr$ | $24dr$ | $2bsd + 2bsr$ |
| LoRA-FA | $dr$ | $2(d^2 + 2dr)$ | $2dr$ | $12dr$ | $2bsr$ |

tween them. For MHA and FFN, layernorm and residual connection are applied to improve model performance.

## 2.2 LOW-RANK ADAPTATION

As fine-tuning LLMs with full parameter is very expensive, parameter-efficient fine-tuning (PEFT) methods particularly LoRA (Hu et al., 2022) are proposed to update only a small fraction of model parameters to alleviate the memory overhead, while achieving comparable performance in fine-tuning LLMs with Full-FT. Specifically, LoRA adds a low-rank adaptor besides the weight of a linear layer as follows:

$$Y = XW + \frac{\alpha}{r}XAB, \tag{1}$$

where $W \in \mathbb{R}^{d_{in} \times d_{out}}$ is the pre-trained weight, $d_{in}$ is the input dimension, and $d_{out}$ is the output dimension. We omit the bias term as it does not affect our analysis. $X \in \mathbb{R}^{b \times s \times d_{in}}$ and $Y \in \mathbb{R}^{b \times s \times d_{out}}$ are input and output tensors, respectively, $b$ is the batch size and $s$ is the sequence length. For the LoRA part, $A \in \mathbb{R}^{d_{in} \times r}$ is the projection-down weight, and $B \in \mathbb{R}^{r \times d_{out}}$ is the projection-up weight, $r$ is the rank size, and $\alpha > 0$ is a hyper-parameter (which by default is set as 8).

For a transformer model, such as GPT (Brown et al., 2020), we typically have $d_{in} = d_{out} = d$ for four linear layers in MHA, and $d_{in} = d, d_{out} = 4d$ (or $d_{in} = 4d, d_{out} = d$) for the first (or second) linear layer in FFN [1], where $d$ is the hidden dimension. By default, we add LoRA modules into all linear layers in transformer blocks to enhance the fine-tuning performance (Zhang et al., 2023b).

**Memory complexity.** For Full-FT, we need to update the weight of $W$ in a linear layer, which has $d_{in} \times d_{out}$ elements, and the total number of weight parameters for a GPT-type model is given by $n = 12d^2L$ [2]. For LoRA, we only update two low-rank matrices, having $(d_{in} + d_{out})r$ elements, and the total number of LoRA parameters for a GPT is $n_r = 18drL$. Thus, LoRA can largely reduce the number of trainable parameters if rank size $r$ is much smaller than $d$.

Now consider the 16-bit mixed-precision training setting, full-parameter fine-tuning takes $2n$ bytes for the model weight, and $14n$ bytes for the gradient and optimizer states (32-bit AdamW's states and parameter copy) (Rajbhandari et al., 2020), while LoRA takes $2n$ bytes for the model weight, and $16n_r$ bytes for adaptor related weight, gradient, and optimizer states. It means that LoRA can reduce this part of memory overhead by about 8 times if we have $n_r \ll n$ (or $r \ll d$).

However, the situation is quite different when comparing the activation memory overhead. Full-FT needs to store the input of $X$ to compute the gradient of $W$, while LoRA needs to store the input of $X$ to compute the gradient of $A$, as well as the low-rank input of $XA$ to compute the gradient of $B$. Specifically, LoRA and Full-FT take $14bsdL + 12bsrL$ bytes and $14bsdL$ bytes of activations (in 16-bit), respectively. Besides, both of them consume activation memory in other components such as attention, GeLU, and layernorm (Korthikanti et al., 2023). Therefore, LoRA is unable to reduce (and even increase) the activation memory cost compared to Full-FT, which unfortunately becomes a new memory bottleneck. A comparison of memory complexity is shown in Table 1.

**Challenges of reducing activation memory.** There are two main ways to reduce the activation memory cost of LoRA. First, one can only fine-tune a small number of linear layers of the model instead of all with LoRA, such as query and value projections in a Transformer model (Hu et al.,

---

[1] The expand dimension is $8d/3$ for LLaMA models by using SwiGLU function (Touvron et al., 2023a).

[2] The total number of full parameters shall be larger as we do not include embeddings and biases for ease-of-presentation.

2022). In this way, other frozen linear layers without LoRA do not need to store their input activations. However, this method could easily sacrifice the fine-tuning task performance (Dettmers et al., 2023), and it also introduces the difficulty of selecting which layers to fine-tune with LoRA (Zhang et al., 2023b). Second, activation recomputation (Chen et al., 2016; Korthikanti et al., 2023) has been proposed to checkpoint only the input of each transformer block during the feed-forward pass, and recompute other necessary activations starting from this checkpoint during the back-propagation pass. However, activation recomputation has very expensive computational cost, which introduces an extra feed-forward pass and takes around $1/3$ of total computing flops (Korthikanti et al., 2023). In this work, we propose to eliminate the requirement of storing input activations of LoRA with no impact on accuracy and efficiency.

## 3 LoRA-FA Method

First of all, we present the design of LoRA-FA method, interpret it from a low-rank model adaptation perspective, and analyze its benefit in reducing the memory overhead. Second, we show that LoRA-FA can be integrated into other memory optimization techniques to enhance its utilization. Third, we discuss the relation between LoRA-FA and gradient compression.

### 3.1 LoRA-FA: LoRA with Frozen Matrix $A$

The LoRA method updates two low-rank matrices $A$ and $B$, and uses $AB$ as the change of a pre-trained and frozen weight $W$ of a linear layer as shown in Eq. (1). As $A$ should be trained during fine-tuning, $X \in \mathbb{R}^{b \times d}$ should be stored during feed-forward for the usage of computing the gradient of $A$ during back-propagation, which thus takes extremely high memory consumption.

To eliminate the requirement of storing $X$, we propose to freeze $A$ in the LoRA layer as shown in Figure 1(c) and we call the new method as LoRA-FA. It means that the input $X$ is always projected to the same space by the fixed matrix of $A$. Formally, the LoRA formula of Eq. (1) is updated as

$$Y = XW + \frac{\alpha}{r}\hat{X}B, \tag{2}$$

where $\hat{X} = XA \in \mathbb{R}^{b \times s \times r}$. Therefore, LoRA-FA only needs to update $B$ so that only $\hat{X}$ needs to be stored during feed-forward for computing the gradient of $B$ during back-propagation. Impressively, LoRA-FA does not change feed-forward and back-propagation computations of LoRA (except skipping the gradient calculation of $A$). Thus, LoRA-FA would not increase the computational overhead during the fine-tuning phase. During the inference, similar to LoRA, it can merge low-rank weights by adding $AB$ into $W$, introducing no extra inference latency compared to a fully fine-tuned model.

**Low-rank model adaptation.** In LoRA-FA, the change of weight during model adaptation is constrained in a low-rank space as follows:

$$\Delta W = AB = Q\bar{B} = \sum_{i=1}^{r} Q_{:,i}\bar{B}_{i,:}, \tag{3}$$

where $A = QR$ is the QR decomposition, and the $r$ columns of $Q$, i.e., $Q_{:,i}$ for $i = 1, \cdots, r$, are orthogonal unit vectors, when $A$ is a rank-$r$ matrix which can be constructed during initialization. We denote $\bar{B} = RB$, and derive that $\Delta W_{:,j} = \sum_{i=1}^{r} Q_{:,i}\bar{B}_{ij}$, so any column of $\Delta W$ is a combination of $k$ orthogonal vectors. Thus, the change of weight in LoRA-FA resides in a low-rank space as like LoRA, but it is just constrained in the column space of $A$. We will show such a constraint in LoRA-FA would not sacrifice fine-tuning performance with extensive experiments in §4.

**Memory complexity.** We study the memory complexity of LoRA-FA in details. For a LoRA-FA module, it only computes the gradient of $B$, which has $d_{out} \times r$ elements. In a GPT-type model, the total number of trainable parameters is $n_r/2 = 9drL$, i.e., half the amount of trainable parameters in LoRA. Thus, the memory cost for model weight and adaptor related states is $2n + 9n_r$ bytes in 16-bit mixed-precision training. More importantly, in terms of activation memory, LoRA-FA only stores the low-rank input of $\hat{X} = XA$ to calculate the gradient of $B$, which takes $12bsrL$ bytes of activations (in 16-bit) for all LoRA-FA modules applied to a GPT-type model. Compared to Full-FT, LoRA-FA is memory-efficient by significantly reducing the amount of trainable parameters and input activations as shown in Table 1.

### 3.2 COMBINATION WITH MEMORY OPTIMIZATIONS

LoRA-FA can be naturally combined with advanced memory optimization approaches including weight quantization like QLoRA (Dettmers et al., 2023), weight sharding like ZeRO (Rajbhandari et al., 2020), and selective activation recomputation like FlashAttention (Dao et al., 2022).

**Weight quantization.** As discussed before, the memory cost for model weight in 16-bit format is $2n$, where $n$ is the number of model parameters. For example, the model weight memory cost is 130GB for a LLaMA-65B model, which cannot be held in one NVIDIA A100 (80GB) GPU. In LoRA-FA, as the model weights are frozen during fine-tuning, we can quantize them into lower bit width to reduce the model weight memory overhead without affecting the fine-tuning performance. For example, 8-bit (Dettmers et al., 2022a) and 4-bit quantization methods (Dettmers et al., 2023) can be combined with LoRA-FA to reduce the model weight memory by 2 and even 4 times.

**Weight sharding.** When training a LLM on multiple GPUs with data parallelism, weight sharding or ZeRO stage-3 (Rajbhandari et al., 2020) technique can be combined with LoRA-FA to shard the model weight into different GPUs, so that the per-GPU memory cost is reduced by the number of GPUs. Different from using ZeRO stage-3 in full-parameter fine-tuning, we only shard the model weights and all-gather them to support the feed-forward and back-propagation computations, without sharding the adaptor related weights and their gradients and optimizer states. However, weight sharding has introduced expensive weight gathering communication cost in LoRA-FA, while data parallelism only communicates a small amount of gradients for trainable parameters.

**Selective activation recomputation.** The activation memory overhead exists in other components of a transformer model, such as attention, layernorm, GeLU, and dropout (Korthikanti et al., 2023). To address it, we can use full activation recomputation to store the input of each transformer block. However, it will disable the memory advantage of LoRA-FA over LoRA, as there is no need to store the inputs of LoRA layers with full activation recomputation. To balance the activation cost and recomputation cost, we instead use selective activation recomputation to recompute only a fraction of model components. For example, FlashAttention (Dao et al., 2022) can eliminate the memory cost of attention softmax outputs and accelerate the attention computations with less HBM accesses. Besides, we can recompute the dropout by storing the random generator state to get the exact mask.

### 3.3 RELATION TO GRADIENT COMPRESSION

We discuss the relation between LoRA-FA and low-rank gradient compression (Vogels et al., 2019; Zhang et al., 2023a) which has been effective in reducing communication costs of distributed training Song et al. (2023). Given a LoRA-FA layer (we omit $\alpha/r$ for simplicity), i.e., $Y = XW + XAB$, the gradient of $B$ is computed by

$$dB = A^T X^T dY = A^T dW. \tag{4}$$

The change of $B$ with one update step of vanilla SGD is $\Delta B = -\eta dB$, where $\eta$ is the learning rate, so the change of $W$ with frozen $A$ is $\Delta W = A\Delta B = -\eta AA^T dW$, and $dW$ is the gradient of $W$.

This implies that LoRA-FA is equivalent to a low-rank gradient compression method for Full-FT, in which the calculated weight gradient is compressed by $A^T dW$ (to reduce the gradient communication overhead in the context of gradient compression), and then it is uncompressed by $A(A^T dW)$. Because $A$ is initialized from a normal distribution, we have $\mathbb{E}[AA^T dW] = \mathbb{E}[AA^T]dW = rdW$, which (almost) gives an unbiased gradient compression.

However, the gradient compression has no advantages over LoRA-FA for fine-tuning LLMs, because it still updates the full parameter with large memory overhead, while LoRA-FA with a small amount of trainable weights can also enjoy the reduced gradient communication in a data parallelism setting. Besides, these two methods are slightly different when applying adaptive methods such as AdamW.

## 4 EXPERIMENTS

We report fine-tuning performance and GPU memory usage of LoRA-FA across different model types and tasks. Due to the page limit, more experimental results are given in Appendix.

## 4.1 FINE-TUNING PERFORMANCE

We evaluate the fine-tuning performance of three key approaches: full-parameter fine-tuning (Full-FT), LoRA (Hu et al., 2022) and LoRA-FA on different model types (covering encoder-only, decoder-only, and encoder-decoder models) and model scales (from millions to billions of parameters). Our experiments cover a wide range of tasks, from natural language understanding (NLU) to machine translation (MT) and natural language generation (NLG). Specifically, we evaluate on the GLUE (Wang et al., 2019) benchmark for RoBERTa-base and RoBERTa-large models (Liu et al., 2019), the WMT16 En-Ro translation for T5-small, T5-base, and T5-large models (Raffel et al., 2020), and the MMLU (Hendrycks et al., 2021) for LLaMA models (Touvron et al., 2023a). We follow the setups of prior works (Hu et al., 2022; Dettmers et al., 2023), and we conduct hyper-parameters tuning for each experiment, including the learning rate of $\eta$ in $\{5 \times 10^{-5}, 6 \times 10^{-5}, \ldots, 1 \times 10^{-4}, 2 \times 10^{-4}, \ldots, 5 \times 10^{-3}\}$, and the LoRA rank of $r$ in $\{1, 2, 4, 8, 16, 32, 64, 128\}$. We report the best performance for a fair comparison, and the impact of hyper-parameters has been studied in Appendix B.4. We conduct our experiments on different GPUs: NVIDIA Turing RTX2080Ti for small-sized RoBERTa models, NVIDIA Ada RTX4090 for medium-sized T5 models, and NVIDIA Ampere A100 for large-sized LLaMA models.

### 4.1.1 ROBERTA BASE/LARGE

The RoBERTa (Liu et al., 2019) is an encoder-only model built on BERT (Devlin et al., 2019). We take the pre-trained RoBERTa-base with 125 million parameters and RoBERTa-large with 355 million parameters to evaluate the fine-tuning performance on GLUE. We mainly replicate the result of Transformers (Wolf et al., 2020) and (Dettmers et al., 2023) according to their setup. Before conducting all the experiments, we first carry out a hyper-parameter search on a signal MRPC task to get optimal hyper-parameter settings, which are used in other experiments. We use the batch size of 64 for fine-tuning RoBERTa-base, and the batch size of 32 for fine-tuning RoBERTa-large. We use the sequence length of 128 for fine-tuning both models. We also give the performance of prompt tuning (Lester et al., 2021) for a comprehensive comparison. The result is presented in Table 2.

Table 2: Fine-tuning RoBERTa-base (R-B) and RoBERTa-large (R-L) models with full-parameter fine-tuning (FT), prompt tuning (PT), LoRA (Lo) and LoRA-FA (Lo-FA) on the GLUE benchmark. We use the batch size of 64 for fine-tuning RoBERTa-base, and the batch size of 32 for fine-tuning RoBERTa-large. We set the sequence length to 128 for both models. This result report the best performance when rank is 8. We report the Matthews correlation for COLA, Pearson correlation for STS-B, averaged matched and mismatched accuracy for MNLI, and accuracy for other tasks. Higher is better for all metrics. We also report the number of trainable parameters (# TPs) for each method. We report the average performance and its std (the bottom-right number) in three independent runs for each task.

| Model & Method | # TPs | MRPC | COLA | QNLI | RTE | SST-2 | STS-B | MNLI | QQP | Avg. |
|---|---|---|---|---|---|---|---|---|---|---|
| R-B (FT) | 118.9M | $\mathbf{90.1_{0.3}}$ | $60_{1.1}$ | $\mathbf{92.5_{0.6}}$ | $77.1_{1.4}$ | $\mathbf{94.8_{0.2}}$ | $89.4_{0.1}$ | $\mathbf{87.4_{0.0}}$ | $\mathbf{90.5_{0.0}}$ | $85.2_{0.5}$ |
| R-B (PT) | 1.2M | $71.6_{0.5}$ | $55.2_{0.3}$ | $70.1_{0.6}$ | $47_{1.1}$ | $72.2_{0.3}$ | $68.9_{0.2}$ | $60.2_{0.1}$ | $60_{0.0}$ | $50.4_{0.4}$ |
| R-B (Lo) | 2.4M | $89.5_{0.4}$ | $\mathbf{63.6_{1.3}}$ | $90.5_{0.5}$ | $77.5_{1.6}$ | $94_{0.2}$ | $\mathbf{90.1_{0.1}}$ | $86.8_{0.0}$ | $89.8_{0.1}$ | $85.2_{0.5}$ |
| R-B (Lo-FA) | 1.8M | $90_{0.4}$ | $\mathbf{63.6_{1.6}}$ | $\mathbf{92.5_{0.7}}$ | $\mathbf{77.9_{1.7}}$ | $\mathbf{94.8_{0.2}}$ | $89.6_{0.2}$ | $86.8_{0.0}$ | $90.1_{0.0}$ | $\mathbf{85.7_{0.6}}$ |
| R-L (FT) | 338.9M | $90.1_{0.2}$ | $67.8_{1.0}$ | $94.2_{0.4}$ | $86_{1.4}$ | $96.1_{0.1}$ | $\mathbf{92_{0.1}}$ | $\mathbf{90.2_{0.0}}$ | $\mathbf{91.1_{0.0}}$ | $88.4_{0.4}$ |
| R-L (PT) | 2.1M | $76.4_{0.3}$ | $55_{0.5}$ | $71.2_{0.5}$ | $61.6_{1.3}$ | $79.5_{0.2}$ | $75.9_{0.1}$ | $70.2_{0.0}$ | $71_{0.1}$ | $70.1_{0.4}$ |
| R-L (Lo) | 5.4M | $\mathbf{90.2_{0.2}}$ | $\mathbf{68_{1.2}}$ | $\mathbf{94.4_{0.5}}$ | $\mathbf{86.3_{1.4}}$ | $\mathbf{96.2_{0.2}}$ | $91.9_{0.1}$ | $90_{0.0}$ | $\mathbf{91.1_{0.0}}$ | $\mathbf{88.5_{0.5}}$ |
| R-L (Lo-FA) | 3.7M | $90_{0.2}$ | $\mathbf{68_{1.2}}$ | $\mathbf{94.4_{0.6}}$ | $86.1_{1.5}$ | $96_{0.2}$ | $\mathbf{92_{0.1}}$ | $90.1_{0.0}$ | $\mathbf{91.1_{0.0}}$ | $\mathbf{88.5_{0.5}}$ |

Table 2 shows that both LoRA and LoRA-FA have a much smaller number of trainable parameters than Full-FT, with LoRA-FA further reducing the scale. For example, LoRA-FA takes only 1.5% of full parameters in fine-tuning RoBERTa-base, while LoRA takes 2%. Among all the four methods, prompt tuning has the smallest parameter group yet it failed to meet the desired target. With a rather small scale of parameter group, LoRA-FA can still achieve close (and even better) results of Full-FT.

Specifically, LoRA-FA leads the best on COLA, QNLI, RTE, SST-2 when fine-tuning RoBERTa-base, and on COLA, QNLI, STS-B, QQP when fine-tuning RoBERTa-large. LoRA-FA gets an average accuracy of 85.7% in fine-tuning RoBERTa-base, and 88.5% in fine-tuning RoBERTa-large, which proves that LoRA-FA is capable of fine-tuning RoBERTa on multiple tasks. We also study the convergence performance of LoRA-FA in Appendix B.2, which shows that LoRA-FA has a similar convergence speed compared to LoRA.

### 4.1.2 T5 SMALL/BASE/LARGE

T5 (Raffel et al., 2020) is an encoder-decoder model pre-trained on a multi-task mixture of unsupervised and supervised tasks. T5 works well on various of tasks by prepending a prefix to the input corresponding to each task. T5 comes in 5 sizes: T5-small (60M), T5-base (220M), T5-large (770M), T5-3B, T5-11B. We use the size of small, base, and large of T5 models to evaluate the performance of Full-FT, LoRA and LoRA-FA on WMT16 En-Ro translation tasks. The evaluation metric includes BLEU and RougeL. We use the batch size of 64 for fine-tuning T5-small, 32 for fine-tuning T5-base, and 16 for fine-tuning T5-large. The result is presented in Table 3.

Table 3: Fine-tuning T5-small, T5-base, and T5-large models with three approaches on the WMT16 En-Ro dataset. This result reports the best performance when rank is 8 for T5-small and T5-base, 16 for T5-large. We report BLEU and ROUGE-L scores. Higher is better for all metrics.

| Model & Method | # Trainable Parameters | BLEU | ROUGE-L |
|---|---|---|---|
| T5-small (Full-FT) | 57.7M | **28.7** | **40.1** |
| T5-small (LoRA) | 0.4M | 27 | 39.6 |
| T5-small (LoRA-FA) | 0.2M | 27 | 39.7 |
| T5-base (Full-FT) | 212.6M | 33.4 | 42.6 |
| T5-base (LoRA) | 1.3M | 32.8 | **43.2** |
| T5-base (LoRA-FA) | 0.6M | **33.5** | 42.8 |
| T5-large (Full-FT) | 703.5M | 36.9 | 49 |
| T5-large (LoRA) | 4.5M | **37** | **49.1** |
| T5-large (LoRA-FA) | 2.25M | **37** | 49 |

As indicated in Table 3, when compared to Full-FT, LoRA-FA exhibits a reduced trainable parameter size. This reduction is quantified as $0.35\%$ for T5-small, $0.28\%$ for T5-base, and $0.32\%$ for T5-large, respectively. LoRA-FA has the smallest requirement of trainable parameters achieving almost the same model performance in both T5-base and T5-large. It shows that LoRA-FA is suitable for fine-tuning relatively large T5 models such as T5-base and T5-large. Besides, LoRA-FA can achieve the same performance in fine-tuning a small model such as T5-small compared to LoRA, but both of them perform slightly worse than Full-FT. This may be because LoRA and LoRA-FA's small parameter group could not handle the fine-tuning dataset when applied to a small size base model, e.g., T5-small who has only 57.7M parameters. There is more of interest to apply LoRA-FA into large models.

### 4.1.3 LLAMA

The fine-tuning result on RoBERTa and T5 illustrate that LoRA-FA can be a competitive alternative to Full-FT and LoRA on NLU and MT tasks. We further evaluate if LoRA-FA still prevails on larger decoder-based NLG models, such as LLaMA-7B (Touvron et al., 2023a). The LLaMA models come in sizes ranging from 7B to 65B parameters and were trained on between 1T and 1.4T tokens, establishing the basis of language understanding and generation abilities. To make them follow instructions, we fine-tune a LLaMA-7B model on Alpaca (Taori et al., 2023) and FLANv2 (Wei et al., 2021) instruction datasets. Due to the massive scale of FLANv2, we sample a split that contains 50k training data according to (Dettmers et al., 2023). This split keeps a similar size with Alpaca (52k training samples). Following prior works (Touvron et al., 2023a; Dettmers et al., 2023), we evaluate the average 5-shot performance of fine-tuned LLaMA models on MMLU benchmarks

(Hendrycks et al., 2021), which covers 57 tasks including elementary mathematics, US history, computer science, law, etc. We use the batch size of 32 for fine-tuning LLaMA-7B. The result is presented in Table 4.

Table 4: Average 5-shot MMLU accuracy comparison for LLaMA-7B models fine-tuned with four approaches on two different datasets: Alpaca and FLAN v2. In these four approaches, LoRA-qv only apply low rank adapters to query and value layers. Higher is better for accuracy metric. This result reports the best performance when rank is 64. We report the absolute performance improvements over the base LLaMA-7B model in parentheses.

| Model & Method | # Trainable Parameters | 5-shot MMLU Accuracy |
|---|---|---|
| LLaMA-7b-Alpaca (Full-FT) | 6426.3M | **37.6** (+2.5) |
| LLaMA-7b-Alpaca (LoRA-qv) | 16.8M | 35.4 (+0.3) |
| LLaMA-7b-Alpaca (LoRA) | 152.5M | 37.2 (+2.1) |
| LLaMA-7b-Alpaca (LoRA-FA) | 83M | 37.4 (+2.3) |
| LLaMA-7b-FLANv2 (Full-FT) | 6426.3M | **45.2** (+10.1) |
| LLaMA-7b-FLANv2 (LoRA-qv) | 16.8M | 37.5 (+2.4) |
| LLaMA-7b-FLANv2 (LoRA) | 152.5M | 43.9 (+8.8) |
| LLaMA-7b-FLANv2 (LoRA-FA) | 83M | 44 (+8.9) |

The Table 4 shows that the parameter group of LoRA-FA takes only $1.3\%$ of Full-FT. Full-FT leads the board for both Alpaca and FLANv2, due to its strength of the largest parameter group. Meanwhile, LoRA-FA achieves the competitive performance using only 83 million trainable parameters, and it performs better than LoRA and LoRA-qv in our evaluations. This result shows that LoRA-FA is capable of fine-tuning LLaMA with much less trainable parameters, and can target similar or even better performance than LoRA. The result of LoRA-qv also suggests that it is necessary to apply LoRA to all linear layers. Furthermore, the absolute performance improvements (e.g., $10.1\%$ with FLANv2) over base model validate the importance of instruction tuning, and FLANv2 is more useful than Alpaca to improve the problem solving power of LLMs.

## 4.2 MEMORY EFFICIENCY

LoRA-FA can save a considerable amount of GPU memory usage by reducing the number of trainable parameters and the activation memory compared to Full-FT. We hence give an analysis on the GPU memory cost of 3 approaches (Full-FT, LoRA, and LoRA-FA) in fine-tuning RoBERTa-base/large, T5-small/base/large, LLaMA-7B models. For hyper-parameter settings, we use the batch size of 128 for fine-tuning T5-small, 64 for RoBERTa-base, RoBERTa-large and T5-base, 32 for T5-large and 16 for LLaMA-7B. For LoRA and LoRA-FA, we compare the memory usage under the rank size with the best accuracy performance. All memory efficiencies are measured on a single A100 GPU (80GB). The result is present in Table 5.

Table 5 shows that LoRA-FA can significantly reduce the GPU memory usage during fine-tuning. Compared to LoRA, LoRA-FA has an average of 3GB memory saving in fine-tuning RoBERTa-base, 4 to 7GB memory saving in fine-tuning T5-base, T5-large, RoBERTa-large, and 15GB memory saving in fine-tuning LLaMA-7B, while full-parameter fine-tuning has caused out of memory in fine-tuning LLaMA-7B.

**Effects of advanced memory optimizations.** LoRA-FA can reduce more GPU memory usage when combined with other advanced memory optimization techniques. We apply FlashAttention (Dao et al., 2022), QLoRA (Dettmers et al., 2023), ZeRO stage-3 (Rajbhandari et al., 2020) with LoRA-FA, respectively, for fine-tuning the LLaMA2-7B (Touvron et al., 2023b) model. We use the configuration of NF4 with double-quantization for QLoRA, and load model weights as BF16 for other methods. The experiments of QLoRA with LoRA-FA are conducted on a single A100 80GB GPU, while the experiments of LoRA-FA with FlashAttention and ZeRO are scaled to a 8xA100 80GB cluster. We use the batch size from 1 to 32, sequence length of 1024, and rank of 64. The results are

Table 5: The peak GPU memory (Mem) usage in GB of three fine-tuning approaches for fine-tuning RoBERTa, T5 and LLaMA models. We use the batch size of 128 for fine-tuning T5-small, 64 for RoBERTa-base, RoBERTa-large and T5-base, 32 for T5-large and 16 for LLaMA-7B.

| Model & Method | Rank | Mem | Model & Method | Rank | Mem |
|---|---|---|---|---|---|
| RoBERTa-base (Full-FT) | - | 9.6 | RoBERTa-large (Full-FT) | - | 23.1 |
| RoBERTa-base (LoRA) | 8 | 9.2 | RoBERTa-large (LoRA) | 8 | 22.5 |
| RoBERTa-base (LoRA-FA) | 8 | 6.9 | RoBERTa-large (LoRA-FA) | 8 | 15.7 |
| T5-small (Full-FT) | - | 30.7 | T5-base (Full-FT) | - | 35.7 |
| T5-small (LoRA) | 8 | 29.4 | T5-base (LoRA) | 8 | 32.1 |
| T5-small (LoRA-FA) | 8 | 25.5 | T5-base (LoRA-FA) | 8 | 25.3 |
| T5-large (Full-FT) | - | 40.4 | LLaMA-7B (Full-FT) | - | OOM |
| T5-large (LoRA) | 16 | 34.3 | LLaMA-7B (LoRA) | 64 | 76.74 |
| T5-large (LoRA-FA) | 16 | 27.6 | LLaMA-7B (LoRA-FA) | 64 | 61.49 |

(a) more memory optimizations   (b) number of LoRA layers   (c) rank size of LoRA

Figure 2: GPU memory footprint (GB) comparison under (a) different memory optimizations, (b) different numbers of LoRA layers, and (c) different rank sizes.

shown in Figure 2a, which shows that LoRA-FA works well with all optimization techniques, and could further reduce the GPU memory consumption during LLM fine-tuning.

**Effects of the number of LoRA layers.** Next, we deliver an experiment over the relationship between GPU memory consumption and the number (or percentage) of injected LoRA layers on the LLaMA2-7B model. We use the batch size of 8, sequence length of 1024, and rank of 64, and we run experiments on a single A100 40GB GPU. The result is present in Figure 2b. It illustrates that for the same batch size, LoRA-FA consumes less GPU memory than LoRA under different numbers of LoRA layers. As the number of LoRA layers increases, the memory usage of LoRA increases much faster than LoRA-FA. Thus, applying LoRA into all linear layers would cause OOM, while LoRA-FA can be applied into all linear layers safely to ensure the fine-tuning performance.

**Effects of LoRA rank size.** We further give an analysis of the relationship between the GPU memory footprint and the rank of LoRA and LoRA-FA in fine-tuning LLaMA-7B. We use the batch size of 32, max source/target length of 128 and rank from 1 to 128. As shown in Figure 2c, LoRA takes more GPU memory than LoRA-FA among all ranks (similar results are observed in fine-tuning other models), with a average gap of 1GB. The memory footprint remains steady along ranks, which suggests that LoRA-FA can increase rank size painlessly from 1 to 128 to achieve better performance without OOM, where a very large rank (e.g., larger than 128) is rarely used in practice (Hu et al., 2022).

## 5 CONCLUSION

In this work, we proposed a new PEFT method LoRA-FA, which freezes the parameters of matrix $A$ in LoRA to eliminate the requirement of storing the input activation of the LoRA layer. Thus, LoRA-FA requires much less memory footprint by reducing the trainable parameters and the activation memory cost without sacrificing model performance and fine-tuning efficiency. We conducted extensive experiments on three types of popular LLMs (RoBERTa, T5, and LLaMA) with various model scales. Experimental results show that LoRA-FA reduces the memory consumption by up to $4\times$ and $1.4\times$ compared to full-parameter fine-tuning and LoRA without sacrificing the fine-tuning accuracy.

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

## A APPENDIX: RELATED WORK

**Supervised Fine-tuning.** A pre-trained LLM has a good understanding to human corpus. It can generate texts by continuing the input sequence, or generate the word that has been masked inside the sequence, thus it can be used for various tasks after fine-tuning. Adding a prediction head to a encoder-only model(Devlin et al., 2019; Liu et al., 2019; He et al., 2020), then fine-tuning the model on a dataset with human annotations is a common way for NLU tasks, such as COLA(Warstadt et al., 2019), SST2(Socher et al., 2013), MNLI(Kim et al., 2019) from GLUE benchmark(Wang et al., 2019). LLM can also be used for text-to-text tasks such as translation, and the encoder-decoder structure is needed in such models, including T5(Raffel et al., 2020), mT5(Xue et al., 2021), ByT5(Xue et al., 2022), UMT5(Chung et al., 2023), etc. However, these fine-tuning methods mainly focus on one task (i.e., single-task specialization), which means they can't handle various human instructions (i.e., multi-task generalization). To make LLMs follow instructions, they are fine-tuned on one or more instruction datasets, such as Alpaca(Taori et al., 2023), Self-instruct(Wang et al., 2022a), UnnaturalInstruction(Honovich et al., 2022), SuperNaturalInstructions(Wang et al., 2022b), and FLAN v2(Wei et al., 2021), which consist of paired instruction-output values. This fine-tuning method is called instruction tuning. Many recent instruction-tuned models include InstructGPT(Ouyang et al., 2022), LLaMA 2(Touvron et al., 2023b), Guanaco(Dettmers et al., 2023), Vicuna(Chiang et al., 2023), Falcon(Almazrouei et al., 2023), FLAN-T5(Chung et al., 2022), and they have achieved a great performance in understanding general knowledge across a wide variety of fields from Open-LLM Leaderboard(Edward et al., 2023; Gao et al., 2021; Clark et al., 2018; Zellers et al., 2019; Hendrycks et al., 2021; Lin et al., 2022). In our work, we show that LoRA-FA is able to fine-tune different kinds of models, including encoder-only, encoder-decoder and decoder-only models.

**Parameter-Efficient Fine-tuning.** With the plethora of LLMs being released, models get larger and larger, and full-parameter fine-tuning becomes infeasible to train them on consumer hardware. Parameter-Efficient Fine-tuning (PEFT) approaches are meant to address this problem to reduce the number of trainable parameters by various methods while maintaining performance. For example, Prefix tuning(Li & Liang, 2021) adds prefix parameters to the hidden states in every layer of the model. Prompt tuning(Lester et al., 2021; Liu et al., 2021; Gu et al., 2022) uses template to reconstruct prompt, and only updates parameters related to prompt understanding. IA3(Liu et al., 2022) injects learned vectors to the attention and feed-forward modules. BitFit(Ben Zaken et al., 2022) only updates the bias of the model. LoRA(Hu et al., 2022) adds low-rank adapters as a bypass to linear layers. Among all, LoRA is more often employed to fine-tune LLMs for new tasks, and many recent approaches based on LoRA have been proposed. QLoRA(Dettmers et al., 2023) fine-tunes a quantized model with LoRA. ReLoRA(Lialin et al., 2023) applies a warm-up strategy with LoRA for pre-training. LoraHub(Huang et al., 2023) proposes a strategy to automatically construct LoRA modules for a model in fine-tuning with diverse given tasks. GLoRA(Chavan et al., 2023) adds an additional prompt module to the model, and injects vectors to rescale the weight and bias. In contrast, LoRA-FA has shown its strength in memory usage while preserving performance compared to LoRA when fine-tuning LLMs . We will compare to more PEFT approaches in our future work.

**Memory-efficient Training.** To load or train LLMs onto hardware more efficiently and scalably, many memory-efficient training approaches have been proposed. ZeRO(Rajbhandari et al., 2020) partitions the parameters, gradients and optimizer states equally across all GPUs, and each GPU has a single partition which is also referred to as a shard. At the computing stage, each GPU builds up each layer's weight by asking participating GPUs to send the information it's lacking. Similarly, FSDP(Zhao et al., 2023) shards all of these states across data parallel workers, and it can optionally offload the sharded model parameters to CPUs. Activation recomputation(Korthikanti et al., 2023; Jain et al., 2020; Smith et al., 2022), also known as gradient checkpointing, is used to save memory during the forward pass by recomputing intermediate activations just-in-time during the backward pass. Offloading(Ren et al., 2021; Shoeybi et al., 2020) is a technique to offload the weights or states to CPU and only load them to GPU when needed. Quantization(Dettmers et al., 2023; Jacob et al., 2017; Dettmers et al., 2022b) concentrates to quantize the parameters and gradients into low-bit representations, such as 8-bit floating point or integer, 4-bit floating point or integer, or even 1-bit data type. LoRA-FA shows an advantage for reducing trainable parameters and activation memory in fine-tuning LLMs, and it can be combined with above memory-efficient training approaches.

# B APPENDIX: ADDITIONAL EXPERIMENTS

## B.1 RUNTIME MEMORY BREAKDOWN

When fine-tuning LLMs, we observed that LoRA takes even more memory than full-parameter fine-tuning when batch size getting larger. We breakdown the runtime memory in fine-tuning Llama2-7B when batch size is 16, sequence length is 1024, and rank is 64, to figure out the memory usage percentage of different modules, such as optimizer state and activation memory. The result is presented in 3.

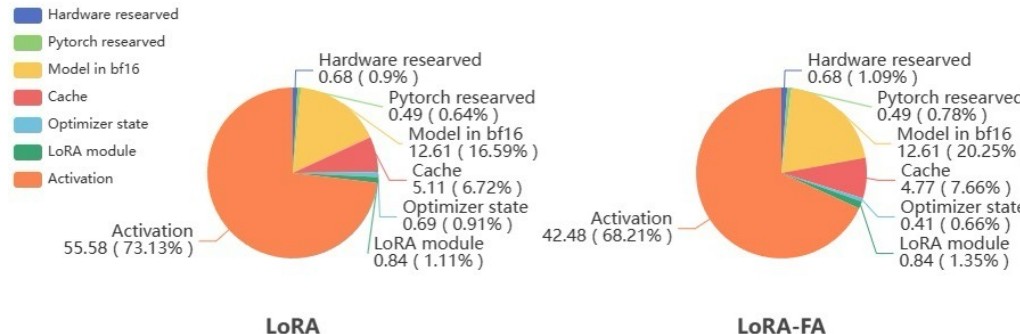

Figure 3: Runtime memory breakdown in fine-tuning Llama2-7B. The batch size is 16, sequence length is 1024, and rank is 64. We report the memory consumption of each module in GB and its percentage.

This result indicates that the activation memory takes the majority of runtime memory in fine-tuning LLMs. LoRA-FA can significantly reduce the activation memory usage compared to LoRA when fine-tuning Llama2-7B (from 55.58 GB to 42.48 GB). This advantage can either help fine-tuning on consumer level GPUs, or can achieve higher throughput by enlarging batch size.

## B.2 CONVERGENCE PERFORMANCE COMPARISON

Due to the less trainable parameters LoRA-FA uses than full-parameter fine-tuning and LoRA, we provide an analysis on convergence performance to see whether it has an impact on convergence speed. We report the convergence results of fine-tuning RoBERT-base on COLA and SST2 datasets as two examples in Figure 4, and the results on other tasks are very similar. The experiments are conducted on 4x A100 40GB, we use a per-device batch size of 320, sequence length of 128 and rank of 8.

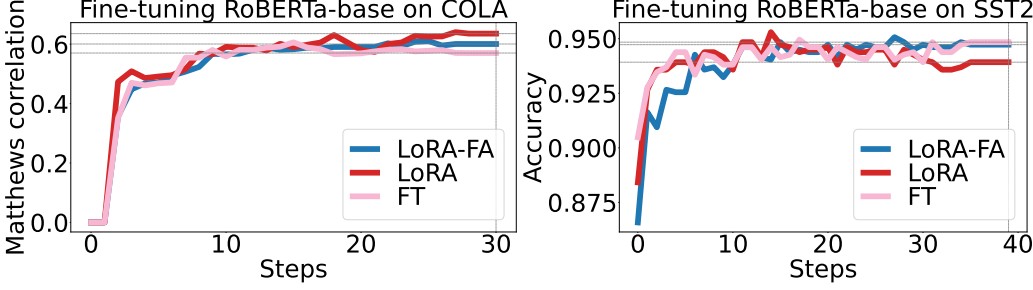

Figure 4: Convergence comparison among full-parameter fine-tuning (FT), LoRA, and LoRA-FA for the RoBERTa-base model on COLA and SST-2 datasets.

Fine-tuning with LoRA-FA does not show any downgrade to convergence speed under suitable hyper-parameter settings according to Figure 4. At the earlier stage (e.g., < 10 steps) of fine-tuning, LoRA and LoRA-FA could converge slower than full-parameter fine-tuning, but they can all reach

the target after several short steps. Overall, our result shows that LoRA-FA has a similar convergence performance compared to full-parameter fine-tuning and LoRA.

## B.3 TIME AND MEMORY EFFICIENCY COMPARISON

We compare the speed and memory footprint between LoRA and LoRA-FA in fine-tuning a LLaMA2-7B (Touvron et al., 2023b) model. We use the batch size from 1 to 32, sequence length of 1024, and rank of 64. We use a single A100 80GB GPU to benchmark all the settings. The results are shown in Figure 5.

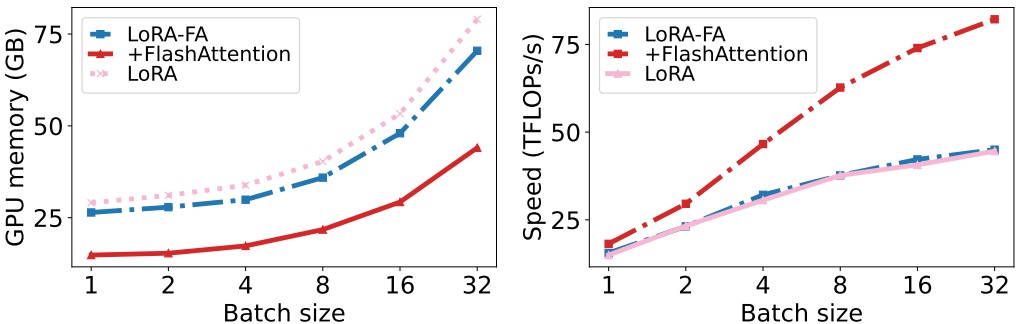

Figure 5: Memory and speed comparison between LoRA, LoRA-FA, and LoRA-FA with FlashAttention.

In terms of speed, LoRA-FA runs slightly faster than LoRA as it does not require to compute the gradient of $A$. In terms of memory footprint, LoRA-FA requires much less peak GPU memory (from 3GB to 8GB) than LoRA by reducing the input activation memory cost as we studied before. In addition, LoRA-FA equipped with FlashAttention (Dao et al., 2022) can further accelerate the training process and significantly reduce the total memory cost.

## B.4 HYPER-PARAMETER STUDIES

### B.4.1 THE EFFECT OF INITIALIZATION OF LOW-RANK MATRIX A

As low-rank weight A is frozen right after being generated, we hence discuss whether different initialization methods will influence the convergence. The LoRA implemented in the PEFT library (Mangrulkar et al., 2022) uses Kaiming uniform, which is also known as He initialization, to generate weight A. Kaiming uniform sampled from $X \sim \mathcal{U}(-bound, bound)$, where

$$bound = \sqrt{\frac{6}{(1 + a^2) \times fan\_mode}} \tag{5}$$

and $a$ is the negative slope of the rectifier used after current layer. We choose 2 other methods in this part for a quick view: all-one initialization to fill A with number 1 and standardized normal initialization to fill A with x sampled from $X \sim \mathcal{N}(0, 1.0)$. We evaluate the performance of fine-tuning RoBERTa-base on MRPC, a task in GLUE benchmark. We use the rank of 8, batch size of 32. The experiment is conducted on a 8x A100 80GB GPU cluster. The result is presented in Table 6.

From Table 6, it shows that initializing the low-rank matrix A with different methods could still perform well in the fine-tuning task if using a suitable hyper-parameter setting. The result indicates that LoRA-FA is robust to various of initialization methods. We adopt Kaiming uniform to implement our LoRA-FA algorithm on top of the PEFT library (Mangrulkar et al., 2022), as it performs slightly better than other methods in this task.

Table 6: Fine-tuning performance of LoRA-FA under different initialization methods.

| Method | Range of $\eta$ | Accuracy |
|---|---|---|
| All-one | $\{1e\text{-}4, \ldots, 9e\text{-}4\}$ | $89.72_{0.4}$ |
| Std-normal | $\{1e\text{-}5, \ldots, 9e\text{-}5\}$ | $88.94_{0.6}$ |
| Kaiming uniform | $\{1e\text{-}5, \ldots, 9e\text{-}5\}$ | $\mathbf{90.00_{0.4}}$ |

### B.4.2 THE EFFECT OF LoRA $\alpha$

LoRA $\alpha$ serves as a control parameter for the scaling coefficient, i.e.,

$$scaling = \frac{\alpha}{r}, \tag{6}$$

where this scaling value is applied to the output of low-rank $B$ projection.

Table 7: Fine-tuning performance of RoBERTa-base on MRPC under different $\alpha$.

| $\alpha$ | Range of $\eta$ | Accuracy | $\alpha$ | Range of $\eta$ | Accuracy |
|---|---|---|---|---|---|
| 1 | $>$1e-3 | 73.12 | 16 | $\{8e\text{-}4, \ldots, 1e\text{-}3\}$ | 88.24 |
| 2 | $>$1e-3 | 78.87 | 32 | $\{4e\text{-}4, \ldots, 9e\text{-}4\}$ | **90.0** |
| 4 | $\{1e\text{-}3, \ldots, 9e\text{-}3\}$ | 82.24 | 64 | $\{2e\text{-}4, \ldots, 6e\text{-}4\}$ | 88.72 |
| 8 | $\{1e\text{-}3, \ldots, 3e\text{-}3\}$ | 86.82 | 128 | $\{1e\text{-}4, \ldots, 5e\text{-}4\}$ | 88.6 |

We conduct the hyper-parameter sensitivity study on $\alpha$, ranging from 1 to 128. Specifically, we evaluate the performance in fine-tuning RoBERTa-base on MRPC, one of the tasks in GLUE. Given the LoRA rank size of $r = 8$, as shown in Table 7, we find that the effective range of $\eta$ for model convergence decreases as $\alpha$ increases, and the optimal $\alpha$ is 32 (which means the *scaling* value is 4). There are much less choices for $\alpha$ than $\eta$ for hyper-parameter tuning. In practice, we fix the *scaling* as 4 for a small rank size, and reduce it to 2 for a large rank size such as 64 and 128.

### B.4.3 THE EFFECT OF LEARNING RATE AND LoRA RANK

LoRA-FA has shown its power in the performance of fine-tuning LLM according to the result above, as it can achieve the close accuracy in time yet has a good memory efficiency runtime. To validate the robustness to hyper-parameter of LoRA-FA, we further conduct a hyper-parameter study about the correlation between rank $r$ and learning rate $\eta$ on LoRA-FA. We compare the performance of LoRA and LoRA-FA in fine-tuning RoBERTa-base on MRPC under a vast range of ranks and learning rates. We use the batch size of 64, and sequence length of 128. The experiment is conducted on a 4x RTX2080Ti GPU server.

The results are present in Figure 6, demonstrating that LoRA and LoRA-FA exhibit similar hyper-parameter space, while LoRA has a slight wider range when $r$ and $\eta$ are around 2 and 5e-5 simultaneously. Both approaches have shown the same pattern that there is a negative correlation between rank $r$ and learning rate $\eta$ with regard to fine-tuning performance, i.e., when rank is going higher, the learning rate should be appropriately reduced to maintain the performance.

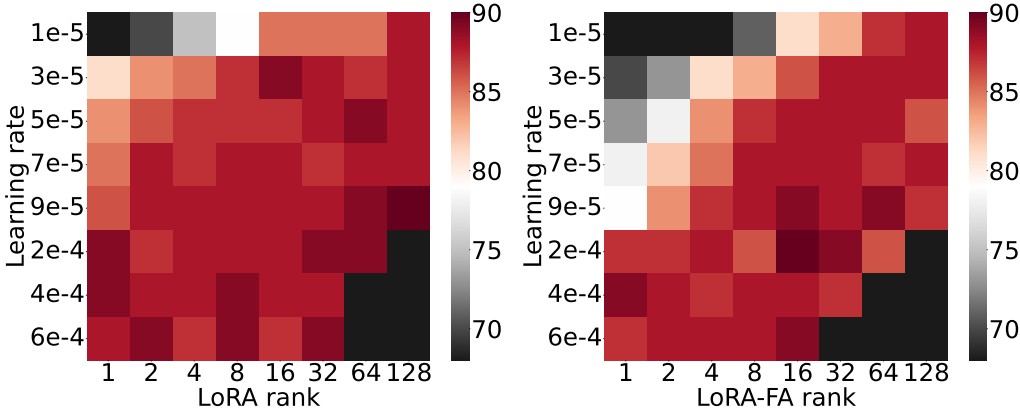

Figure 6: Fine-tuning performance comparison between LoRA and LoRA-FA under different ranks and learning rates for the RoBERTa-base model on the MRPC dataset.

