# OpenReview forum: "LoRA-FA: Memory-efficient Low-rank Adaptation for Large Language Models Fine-tuning"
_ICLR.cc/2024/Conference — Submitted to ICLR 2024_

### Official Review · Reviewer_wXxt · 2023-10-13

**Soundness:** 3 good
**Presentation:** 2 fair
**Contribution:** 3 good
**Rating:** 6
**Confidence:** 5

**Summary:**

This paper provides a minor but meaningful tweak to LoRA, by freezing the A matrix, which has notable benefits in improving training memory. The paper provides both theoretical and empirical justifications for why LoRA-FA achieves close to LoRA performance.

**Strengths:**

- The method is simple, well-explained, and well-justified
- The experiments are fairly comprehensive (though as detailed in weaknesses, still leave some crucial questions unanswered)

**Weaknesses:**

- The writing of the paper at times leans towards over-claiming or making unsuitable comparisons to bolster their method. For example, in section 1 and in other spots, the memory of LoRA-FA is compared to full fine-tuning but not LoRA. Given that LoRA-FA is fundamentally a tweak of LoRA, the natural comparison should be to LoRA, not the full fine-tuning just to make the numbers look more impressive. To the authors: the benefits of LoRA-FA are moderate but clear; there is not need to oversell the method.
- The crux of the evaluation lies in whether LoRA-FA underperforms LoRA, or performs comparably with reduced memory consumption. While the evaluations in the paper are quite comprehensive (spanning 3 model families), I think the current experiment still fall short of resoundingly answering this crucial question, and I would like to see the authors run a set of experiments to address this. To put this more explicitly: to determine if LoRA-FA underperforms LoRA (or not), we need a setting where LoRA is "capacity-constrained", to determine if LoRA-FA has even less "capacity" than LoRA. To do this, we need a setting where LoRA meaningfully underperforms full fine-tuning. For both the RoBERTa and T5 experiments, this is not the case. The LLaMA-7B experiments on Alpaca/FLAN -> MMLU come the closest to this, but the margin is still too small to tell (and essentially no gap at all in the case of Alpaca) (more broadly, MMLU is not a fine measure of LM performance). I can recommend the authors run a set of experiments on something like Super-NaturalInstructions, where there is likely to be a bigger gap (since the evaluation is performed on generated sequences rather than simple multiple-choice knowledge questions).

**Questions:**

My questions are detailed in the weaknesses section above (testing exactly where/how much capacity is lost between LoRA and LoRA-FA). I am willing to update my score given more experiments addressing my question.

---

> ### Author Response · Authors · 2023-11-20
>
> We appreciate the positive and valuable comments.
>
> **Limitation 1**: The writing of the paper at times leans towards over-claiming or making unsuitable comparisons to bolster their method.
>
> **Response**: Thank you for your insightful comment, we will refactor the paper to make it more impartial.
>
> **Limitation 2**: The crux of the evaluation lies in whether LoRA-FA underperforms LoRA, or performs comparably with reduced memory consumption. While the evaluations in the paper are quite comprehensive (spanning 3 model families), I think the current experiment still fall short of resoundingly answering this crucial question, and I would like to see the authors run a set of experiments to address this.
>
> **Response**: Thank you for your kind guidance. To address this problem, we conduct the  Llama2-7B fine-tuning on Super-NaturalInstructions dataset. Super-NaturalInstructions is a challenging dataset for LLMs. Specifically, we use the task001_quoref_question_generation.json and task002_quoref_answer_generation.json as the training set, and set the sequence length to 2048, batch size to 4, train epochs to 5. We report eval_loss of 3 learning rates: [7e-5, 1e-4, 4e-4] as follows.
>
> | Learning rate | 7e-5 | 1e-4 | 4e-4 |
> |---------------|------|------|------|
> | LoRA          | 0.45 | 0.53 | 4.27 |
> | LoRA-FA       | 0.45 | 0.47 | 0.51 |
>
> In the results, both LoRA and LoRA-FA can achieve similar eval_loss. When $lr=4e-4$, LoRA didn't perform well because the learning rate is too large for LoRA. Overall, LoRA-FA shows strong capability similar to LoRA when fine-tuning on generalization tasks.

---

> > ### Comment · Reviewer_wXxt · 2023-11-22
> >
> > - As mentioned in my original comment, the experiment needs to be isolate a case where LoRA underperforms full fine-tuning. As such, we will also need full fine-tuning results to compare to.
> > - I don't believe this is the standard application of the Super-NaturalInstructions dataset. The S-NI dataset is intended to be a large multi-task dataset, with many training tasks, and evaluation on separately held-out tasks, to determine the generalization capability of a model. As such, fine-tuning on just two tasks (and it is not clear what the evaluation tasks here are) isn't quite the appropriate setup.

---

### Official Review · Reviewer_XvzM · 2023-10-31

**Soundness:** 3 good
**Presentation:** 3 good
**Contribution:** 2 fair
**Rating:** 5
**Confidence:** 4

**Summary:**

In this paper, the authors introduce to reduce for activation memory during fine-tuning, called LoRA-FA. LoRA-FA introduces a memory-efficient approach by freezing certain weights of LoRA layers, significantly reducing activation memory without compromising performance or incurring additional computational costs. Experiments across various model types and scales, including RoBERTa, T5, and LLaMA, demonstrate that LoRA-FA consistently maintains fine-tuning accuracy while reducing overall memory costs by up to 4x and 1.4x compared to full-parameter fine-tuning and LoRA, respectively. Additionally, LoRA-FA is compatible with advanced memory optimization methods like FlashAttention, QLoRA, and ZeRO.

**Strengths:**

* When comparing activation memory among various existing fine-tuning methodologies, it is evident that LoRA does not significantly reduce activation memory compared to full fine-tuning.
* In this paper, the authors propose a simple yet effective approach to reduce activation memory size during the fine-tuning phase by selectively freezing certain portions of the LoRA adaptation layer.
* The experimental findings demonstrate that the proposed LoRA-FA achieves performance comparable to LoRA across various large language models (LLMs), including LLaMA, T5, and RoBERTa, in downstream tasks.
* The study highlights the compatibility of LoRA-FA with advanced memory optimization techniques such as FlashAttention, QLoRA, and ZeRO.

**Weaknesses:**

* The discussion on the benefits of reduced activation memory through LoRA-FA is lacking. In the finetuning phase, unlike the inference phase, both sequence length and batch size are longer, resulting in high GPU utilization. Therefore, it is not considered practically significant to make LoRA-FA more memory efficient than the existing LoRA.
* I think efficient finetuning becomes more crucial as the model size increases. Therefore, it is necessary to demonstrate that as the model size grows, it shows performance similar to LoRA. However, the models experimented in the paper were limited to sizes up to 7B, which is relatively small. The trends in models larger than 7B remain unknown.
* As I consider LoRA to be a comprehensive methodology that encompasses LoRA-FA, it is difficult to anticipate that LoRA-FA would exhibit better accuracy than LoRA.
* The cost required for inference after finetuning is completed is the same for both LoRA and LoRA-FA.

**Questions:**

* Is there any performance difference based on the initialization method for the LoRA adaptation layer A?
* When integrating the proposed LoRA-FA into QLoRA, how does it impact the performance of CSR or MMLU in models such as LLaMA, RoBERTa, and T5? While Section 4 of the experiment results demonstrates the approach of freezing adaption layer A in various LLMs' LoRA, including QLoRA, there seems to be a mention of potential application without concrete results on the performance after actual finetuning.
* When looking at Figure 4 in the Appendix, it appears that LoRA and LoRA-FA exhibit a similar trend in TFLOPS. What are the benefits gained during the Finetuning phase by actually reducing activation memory?

---

> ### Author Response · Authors · 2023-11-20
>
> We thank the reviewer for the feedback.
>
> **Question 1**: Is there any performance difference based on the initialization method for the LoRA adaptation layer A?
>
> **Response**: We discussed the performance difference of three different initialization methods for the LoRA adaptation layer A in Appendix B.3.1.
>
> **Question 2**: When integrating the proposed LoRA-FA into QLoRA, how does it impact the performance of CSR or MMLU in models such as LLaMA, RoBERTa, and T5? While Section 4 of the experiment results demonstrates the approach of freezing adaption layer A in various LLMs' LoRA, including QLoRA, there seems to be a mention of potential application without concrete results on the performance after actual finetuning.
>
> **Response**: The fine-tuning with QLoRA is still on-going, we will release the result ASAP.
>
> **Question 3**: When looking at Figure 4 in the Appendix, it appears that LoRA and LoRA-FA exhibit a similar trend in TFLOPS. What are the benefits gained during the Finetuning phase by actually reducing activation memory?
>
> **Response**: Thank you for pointing out the issue! We have corrected the results in Table 5, it was because activation recomputation was turned on when conducting memory benchmark experiments. We ensure all of the configures are right and re-conducted the experiment. Specifically, we fine-tuning Llama2-7B with sequence length = 1024. In the new results, we have several findings:
>
> 1. LoRA-FA reduces ~13GB memory compared to LoRA in fine-tuning Llama2-7B when batch size is 16.
> 2. In practice, when fine-tuning Llama2-7B on various hardware, LoRA-FA has better capability than LoRA. For example, on RTX4090-24G, LoRA can only fine-tuning Llama2-7B when $batch size \leq 2$, while LoRA-FA can enlarge the batch size to 4; on A100-40G, LoRA works with batch size = 4, while LoRA-FA works with batch size = 8.
>
> We also give a runtime memory breakdown analysis of fine-tuning Llama2-7B as follows when batch size is 16, to figure out the memory usage percentage of different modules, such as optimizer state and activation memory.
>
> |                    | LoRA        |                | LoRA-FA     |                |
> |:------------------:|:-----------:|:--------------:|:-----------:|:--------------:|
> |                    | Memory (GB) | Percentage (%) | Memory (GB) | Percentage (%) |
> | Hardware researved | 0.7         | 0.9            | 0.7         | 1.1            |
> | Pytorch researved  | 0.5         | 0.6            | 0.5         | 0.8            |
> | Model in bf16      | 12.6        | 17             | 12.6        | 20.3           |
> | LoRA module        | 0.8         | 1.1            | 0.8         | 1.4            |
> | Cache              | 5.1         | 6.7            | 4.8         | 7.7            |
> | Optimizer state    | 0.7         | 0.9            | 0.4         | 0.4            |
> | Activation         | 55.6        | 73.1           | 42.3        | 68.2           |
>
> This result shows that LoRA-FA can significantly reduce the activation memory usage compared to LoRA when fine-tuning Llama2-7B (from 55.6 GB to 42.3 GB). This advantage can either help fine-tuning on consumer level GPUs, or can achieve higher throughput by enlarging batch size.

---

> ### Comment · Reviewer_XvzM · 2023-11-22
>
> Thank you for the detailed answers and results.
>
> Some of my concerns have been addressed, but i still my concern that using LoRA-FA for finetuning can enhance training throughput. I agree with the effectiveness of LoRA-FA in efficiently reducing activation memory, considering it a significant strength of the paper.
>
> However, during the fine-tuning phase, it is common to have significantly larger batch sizes and sequence lengths compared to inference, resulting in an exponential increase in time complexity over space complexity. As a result, fine-tuning becomes bound by computation, and in my view, the proposed method does not effectively reduce computation costs during the fine-tuning process, even though it minimizes activation memory.
>
> As a result, the proposed method, in my understanding, does not reduce computation costs during fine-tuning, even if activation memory is minimized.
>
> Hence, I believe the benefits in terms of throughput during the fine-tuning phase, compared to LoRA, may be limited. I consider the experimental results in Figure 4 in the Appendix as evidence supporting this view.
>
> If there were more convincing explanations for this aspect, I would be willing to reconsider and potentially raise my score. However, lacking sufficient clarity on this matter, I would like to keep my rating.

---

### Official Review · Reviewer_s6YB · 2023-11-01

**Soundness:** 2 fair
**Presentation:** 1 poor
**Contribution:** 2 fair
**Rating:** 5
**Confidence:** 3

**Summary:**

The paper presents LoRA-FA, a new parameter-efficient fine-tuning (PEFT) approach for Large Language Models (LLMs). LoRA-FA is an extension of the LoRA method which minimizes memory usage by freezing the A matrix in LoRA layers. This approach alleviates the need to store the input activation of the LoRA layer, leading to a reduction in memory footprint during fine-tuning. The authors provide comprehensive experimental evidence showcasing the efficacy of LoRA-FA across various LLMs including RoBERTa, T5, and LLaMA. The results demonstrate LoRA-FA's memory savings without compromising on fine-tuning performance.

**Strengths:**

* The proposed LoRA-FA method cuts down GPU memory usage by reducing both the number of trainable parameters and the activation memory compared to traditional full fine-tuning.
* Comprehensive experimental results on diverse models, including RoBERTa, T5, and LLaMA, demonstrate that LoRA-FA maintains competitive accuracy relative to both full fine-tuning and the original LoRA.
* The paper lucidly presents the background of parameter-efficient fine-tuning and proposes an expanded method that inherits the advantages of the previous work, LoRA.

**Weaknesses:**

* The proposed method, which involves freezing the LoRA weight A from the existing LoRA, appears incremental in terms of novelty.
* In the context of "reducing the number of trainable parameters", as mentioned in the LoRA paper, the previously proposed PEFT method significantly reduced the number of trainable parameters. This led to a drastic reduction in the memory usage of the optimizer state, bringing it down to megabytes. Thus, using fewer trainable parameters than LoRA does not yield a significant difference.
* Regarding "reducing the activation memory", while LLaMA uses a max sequence length of 2k, the LLaMA2 model employs a 4k length. It's evident that models are gravitating towards longer sequence lengths. As per Table 1, LoRA-FA utilizes '2bsr' of memory (compared to LoRA's '2bsd+2bsr'), but the advantage in memory savings during training becomes less pronounced as the model's sequence length increases relative to LoRA.
* The proposed methodology primarily targets the reduction of memory usage during training. In section 4.2, Table 5, the variance in memory peak between LoRA and LoRA-FA is minimal, especially for generative models like LLaMA-7B.
* As stated in the LoRA paper, amplifying the trainable parameters doesn't notably affect accuracy (refer to Figure 2). For a more precise evaluation, I suggest the following comparative experiments with LoRA:
    * Match LoRA and LoRA-FA at the same level of trainable parameters and then compare their accuracy and memory peaks.
    * Adjust both LoRA and LoRA-FA to similar peak memory levels and compare their MMLU accuracy.
    * Examine the peak memory usage of both LoRA and LoRA-FA when using generative models larger than LLaMA-7B to determine if the gap increases as the model size grows.

**Questions:**

Covered in the weaknesses section.

---

> ### Author Response · Authors · 2023-11-20
>
> We appreciate the detailed comments from this reviewer.
>
> **Movation**: When fine-tuning LLMs, we observed that LoRA takes even more memory than full-parameter fine-tuning when batch size getting larger. This is due to the activation memory of sub-modules, such as attention or MLP. Inside these sub-modules, the low-rank adapter A has much more activation memory usage than adapter B. We hence propose LoRA-FA to address this problem.
>
> **Limitation 1**: In the context of "reducing the number of trainable parameters", as mentioned in the LoRA paper, the previously proposed PEFT method significantly reduced the number of trainable parameters. This led to a drastic reduction in the memory usage of the optimizer state, bringing it down to megabytes. Thus, using fewer trainable parameters than LoRA does not yield a significant difference.
>
> **Response**: Thank you for your insightful comment. It is true that using fewer trainable parameters than LoRA does not often yield a significant difference in LoRA-like methods. Our intention was not to imply that using fewer trainable parameters than LoRA would reduce more GPU memory. Rather, our emphasis was to show that LoRA-FA reduces great activation memory by freezing the update of low rank adapter A, which can also reduce the trainable parameters. We will ensure to clarify this in our revised manuscript to avoid any misunderstanding.
>
> **Limitation 2**: Regarding "reducing the activation memory", while LLaMA uses a max sequence length of 2k, the LLaMA2 model employs a 4k length. It's evident that models are gravitating towards longer sequence lengths. As per Table 1, LoRA-FA utilizes '2bsr' of memory (compared to LoRA's '2bsd+2bsr'), but the advantage in memory savings during training becomes less pronounced as the model's sequence length increases relative to LoRA.
>
> **Response**: This is a very detailed comment. Regarding to Table 1, in a linear layer, LoRA-FA saves `2bsd` usage compared to LoRA in activation memory, which is linearly related to the coefficient sequence length. This means the memory saving has a linear-like scaling efficiency regarding to sequence length. We further conduct an experiment with longer sequence length in fine-tuning Llama2-13B. Specifically, we set the sequence length from 1024 to 2048, and the batch size to 2. We report the peak GPU memory usage as follows.
>
> | seq_length | 1024  | 2048  |
> |------------|-------|-------|
> | LoRA       | 50 GB | 71 GB |
> | LoRA-FA    | 45 GB | 62 GB |
>
> This result shows that LoRA-FA saves more GPU memory when using longer sequence length.
>
> **Limitation 3**: The proposed methodology primarily targets the reduction of memory usage during training. In section 4.2, Table 5, the variance in memory peak between LoRA and LoRA-FA is minimal, especially for generative models like LLaMA-7B.
>
> **Response**: Thank you for pointing out the issue! We have corrected the results in Table 5, it was because activation recomputation was turned on when conducting memory benchmark experiments. We ensure all of the configures are right and re-conducted the experiment. Specifically, we fine-tuning Llama2-7B with sequence length = 1024. In the new results, we have several findings:
>
> 1. LoRA-FA reduces ~13GB memory compared to LoRA in fine-tuning Llama2-7B when batch size is 16.
> 2. In practice, when fine-tuning Llama2-7B on various hardware, LoRA-FA has better capability than LoRA. For example, on RTX4090-24G, LoRA can only fine-tuning Llama2-7B when $batch size \leq 2$, while LoRA-FA can enlarge the batch size to 4; on A100-40G, LoRA works with batch size = 4, while LoRA-FA works with batch size = 8.
>
> We also give a runtime memory breakdown analysis of fine-tuning Llama2-7B as follows when batch size is 16, to figure out the memory usage percentage of different modules, such as optimizer state and activation memory.
>
> |                    | LoRA        |                | LoRA-FA     |                |
> |:------------------:|:-----------:|:--------------:|:-----------:|:--------------:|
> |                    | Memory (GB) | Percentage (%) | Memory (GB) | Percentage (%) |
> | Hardware researved | 0.7         | 0.9            | 0.7         | 1.1            |
> | Pytorch researved  | 0.5         | 0.6            | 0.5         | 0.8            |
> | Model in bf16      | 12.6        | 17             | 12.6        | 20.3           |
> | LoRA module        | 0.8         | 1.1            | 0.8         | 1.4            |
> | Cache              | 5.1         | 6.7            | 4.8         | 7.7            |
> | Optimizer state    | 0.7         | 0.9            | 0.4         | 0.4            |
> | Activation         | 55.6        | 73.1           | 42.3        | 68.2           |
>
> This result shows that LoRA-FA can significantly reduce the activation memory usage compared to LoRA when fine-tuning Llama2-7B (from 55.6 GB to 42.3 GB). This advantage can either help fine-tuning on consumer level GPUs, or can achieve higher throughput by enlarging batch size.

---

> ### Author Response · Authors · 2023-11-20
>
> **Limitation 4**:
>
> - Match LoRA and LoRA-FA at the same level of trainable parameters and then compare their accuracy and memory peaks.
> - Adjust both LoRA and LoRA-FA to similar peak memory levels and compare their MMLU accuracy.
> - Examine the peak memory usage of both LoRA and LoRA-FA when using generative models larger than LLaMA-7B to determine if the gap increases as the model size grows.
>
> **Response**:
>
> - We set the rank of LoRA to 64, and the rank of LoRA-FA to 128, to maintain the same level of trainable parameters. We fine-tune Llama2-7B on Super-NaturalInstructions dataset with sequence length = 1024. We report the eval_loss of both approaches. From our result, both LoRA and LoRA-FA can achieve a similar performance (eval_loss = 0.45). LoRA-FA saves 13GB of peak GPU memory compared to LoRA. This means that enlarging rank doesn't yield a great difference in memory consumption.
> - Since enlarging rank doesn't yield a great difference in memory consumption, we choose to adjust the batch size of both methods to maintain a similar peak memory levels. From our result, both LoRA and LoRA-FA achieves a similar MMLU accuracy when fine-tuning Llama2-7B on Super-NaturalInstructions dataset (accuracy = ~46.5).
> - We have checked the peak memory usage when fine-tuning Llama2-7B and Llama2-13B using batch size 4, the results didn't show much increase in GPU memory gap as the model size grows.

---

> > ### Comment · Reviewer_s6YB · 2023-11-21
> >
> > Dear author,
> > Thank you for your response to my queries.
> > However, I still have some growing concerns that I would like author to address:
> > * I couldn’t find any results addressing Limitation 4 in author’s response. Could you please direct me to these results?
> > * The experimental configurations of the paper may need revision, as it is difficult to locate rank information in all the accuracy report tables.
> > * In the experiments shown in Table 4 of the paper, it appears that the rank configuration for LoRA-qv is set at 4, and LoRA-FA is applied to all linear layers. Are there results available for LoRA-qv with a higher rank, like 64 or 128? If so, does LoRA-qv still exhibit lower accuracy in these cases?
> > * In your response to Limitation 4, rather than addressing my concerns, it seems they were exacerbated. I believe finding a configuration with similar peak memory and accuracy to LoRA-FA by reducing the layers where LoRA is applied and adjusting the rank should be feasible.
> > * Regarding the experiments where you match LoRA and LoRA-FA at the same level of trainable parameters to compare their accuracy and memory peaks: is there a notable difference in memory peak and accuracy when LoRA-FA is applied to all layers, while LoRA is applied only to the attention layers of LLaMA (excluding gate_proj, up_proj, down_proj) with an adjusted rank? Additionally, if LoRA is selectively applied to the combinations of query, key, value, or output projection (o_proj) in the attention layers, does adjusting the rank still result in LoRA-FA having superior accuracy?
> > * To convincingly demonstrate that LoRA-FA is superior to LoRA, it seems necessary to show instances where LoRA-FA achieves higher accuracy than LoRA under a limited peak memory constraint. For example, in Figure 2-b of the paper, when the peak GPU memory is limited to 38.5, does LoRA-FA still maintain better accuracy than LoRA, even after sweeping through various configurations (rank, layers applied, etc.) of LoRA?
> > * In response of experiments on “Adjusting both LoRA and LoRA-FA to similar peak memory levels and comparing their MMLU accuracy,” author claimed that merely adjusting the rank size does not significantly impact memory consumption. This seems to be an expected result when only the rank size is modified as we already discussed in the Limitation 1. To effectively demonstrate the competitiveness of LoRA-FA, a study showing the effects of reducing the layers where LoRA is applied, thereby lessening the layers calculating gradients and impacting memory consumption, would have been beneficial.
> > * Based on the response to “Examine the peak memory usage of both LoRA and LoRA-FA when using generative models larger than LLaMA-7B to determine if the gap increases as the model size grows,” author reported that the memory gap does not widen as the model size increases. Does this suggest that the efficiency of LoRA-FA diminishes with larger models?

---

> > > ### Author Response · Authors · 2023-11-22
> > >
> > > Dear reviewer,
> > >
> > > We appreciate your kind guidance. The limitations are also our concerns. We conducted more experiments, to demonstrate that LoRA-FA is superior to LoRA. We report the result in the responses first, and then add this result to the paper if there still is time.
> > >
> > > **Limitation 1 & 2**: I couldn't find any results addressing Limitation 4 in author's response. Could you please direct me to these results? The experimental configurations of the paper may need revision, as it is difficult to locate rank information in all the accuracy report tables.
> > >
> > > **Response**: We make our Limitation 4 results clearer in the responses below. We add detailed experiment settings (batch size, rank, etc.) to every table and figures' captions, to avoid any misunderstanding.
> > >
> > > **Limitation 3**: In the experiments shown in Table 4 of the paper, it appears that the rank configuration for LoRA-qv is set at 4, and LoRA-FA is applied to all linear layers.
> > >
> > > **Response**: The experiments in Table 4 report the best accuracy under a hyper-parameters tuning space in fine-tuning LLaMA-7B, where the exact `best rank` is 64 for LoRA, LoRA-qv and LoRA-FA. Similar to Limitation 1 and 2, we will specify the detailed settings in the table’s caption. LoRA-qv has much less trainable parameters than LoRA and LoRA-FA, due to LoRA-qv only applies LoRA to linear layer q and v, where LoRA and LoRA-FA apply LoRA to all linear layers. The results show that LoRA-qv still exhibits lower accuracy even under larger rank (rank=64).

---

> > > ### Author Response · Authors · 2023-11-22
> > >
> > > **Limitation 4**: Adjusting both LoRA and LoRA-FA to the same level, to check their memory and accuracy.
> > >
> > > **Response**: This is the most important part among these responses. We provide detailed explanations to every question.
> > >
> > > - **Q.1: Find a configuration with similar peak memory of LoRA and LoRA-FA.**
> > > - A.1: To find such a configuration, we can either adjust the number of LoRA layers applied, e.g., 50% of LoRA layers are applied to the model; or adjust the position of LoRA layers, e.g., apply LoRA layers only to q and v.
> > >
> > > - **Q.2: Is there a notable difference in memory peak and accuracy when LoRA-FA is applied to all layers, while LoRA is applied only to the attention layers of LLaMA (excluding gate_proj, up_proj, down_proj) with an adjusted rank?**
> > > - A.2: Our configuration for Q.2 is : apply LoRA to q, k, v; apply LoRA-FA to all linear layers, set the sequence length to 2048, and batch size to 16, choose rank from [1,2,4,8,16,32,64,128]. We report the peak memory and MMLU accuracy of fine-tuning Llama2-7B.
> > > We first report the memory in GB under different ranks.
> > >
> > > | rank                  | 1     | 2     | 4     | 8     | 16    | 32    | 64    | 128   |
> > > |-----------------------|-------|-------|-------|-------|-------|-------|-------|-------|
> > > | LoRA (attention only) | 63.32 | 63.47 | 63.51 | 63.58 | 63.7  | 63.97 | 64.44 | 64.86 |
> > > | LoRA-FA (all linear)  | 61.24 | 61.26 | 61.32 | 61.83 | 61.64 | 61.9  | 62.51 | 63.45 |
> > >
> > > This table shows that under this configuration, i.e., when LoRA-FA is applied to all layers, while LoRA is applied only to the attention layers of Llama2, LoRA and LoRA-FA maintain similar peak memory, and the memory doesn't yield a significant difference when enlarging the rank. We then choose rank = 64 (due to time limitation) to fine-tuning Llama2-7B on alpaca, and report the MMLU accuracy. The results are presented below.
> > >
> > > | rank                  | 64   |
> > > |-----------------------|------|
> > > | LoRA (attention only) | 47.4 |
> > > | LoRA-FA (all linear)  | 54.1 |
> > >
> > > This table shows that under resource-constrained scenarios, LoRA-FA can achieve better fine-tuning performance compared to LoRA, even when they consume the same memory.
> > >
> > > - **Q.3: Additionally, if LoRA is selectively applied to the combinations of query, key, value, or output projection (o_proj) in the attention layers, does adjusting the rank still result in LoRA-FA having superior accuracy?**
> > > - A.3: Due to time limitations, we are still benchmarking on this task and will release it in later paper revision. However combine the result of **Q.2** and **Q.4** we may still have a conclusion, to prove that LoRA-FA achieves higher accuracy than LoRA under a limited peak memory constraint.
> > >
> > > - **Q.4: It seems necessary to show instances where LoRA-FA achieves higher accuracy than LoRA under a limited peak memory constraint.**
> > > - A.4: To address this question, and provide a different view from **Q.2**, we use a configuration where the number of LoRA layers are varying. Specifically, we apply LoRA and LoRA-FA to all layers, and set the sequence length to 2048, batch size to 16, rank to 64, and choose the percentage of LoRA layers from 10% to 100%.
> > > We first report the memory in GB under different percentages.
> > >
> > > | percentage            | 10%   | 20%   | 30%   | 40%   | 50%  | 60%   | 70%   | 80%   | 90%   | 100%  |
> > > |-----------------------|-------|-------|-------|-------|------|-------|-------|-------|-------|-------|
> > > | LoRA (attention only) | 59.52 | 60.73 | 61.92 | 64.45 | 66.4 | 67.79 | 69.42 | 71.07 | 73.15 | 75.33 |
> > >
> > > Since the peak memory of LoRA with 30% (61.92 GB) is similar to LoRA-FA in Q.2 (62.51 GB), we then choose this configuration to perform the fine-tuning. The result is presented below.
> > >
> > > |                 | memory (GB) | Accuracy |
> > > |-----------------|-------------|----------|
> > > | LoRA (30%)      | 61.92       | 45.4     |
> > > | LoRA-FA (100%)  | 62.51       | 54.1     |
> > >
> > > This table shows that when adjusting the number of LoRA layers applied, to match a similar peak memory with LoRA-FA, LoRA-FA achieves a better fine-tuning performance compared to LoRA.

---

> > > ### Author Response · Authors · 2023-11-22
> > >
> > > - **Q.5: A study showing the effects of reducing the layers where LoRA is applied, thereby lessening the layers calculating gradients and impacting memory consumption,  would have been beneficial.**
> > > - A.5: Specifically, we apply LoRA and LoRA-FA to all layers, and set the sequence length to 2048, batch size to 16, rank to 64. We report the peak memory in GB for reducing sub-modules of LoRA layers.
> > >
> > > |  Modules | All   | Only attention | Only MLP | no q  | no k  | no v  | no o  | no up | no down | no gate |
> > > |----------|-------|----------------|----------|-------|-------|-------|-------|-------|---------|---------|
> > > | LoRA     | 75.33 | 64.44          | 67.77    | 72.94 | 73.67 | 73.05 | 73.47 | 73.14 | 69.3    | 73.14   |
> > > | LoRA-FA  | 62.51 | 56.72          | 58.85    | 60.55 | 60.57 | 60.35 | 60.38 | 60.43 | 58.89   | 60.4    |
> > >
> > > This result shows that LoRA-FA can always reduce ~13GB memory compared to LoRA, when cutting down specific sub-modules. In each method, cut down single linear doesn't yield great difference.
> > >
> > > - **Q.6: (When fine-tuning Llama2-13B) Does this suggest that the efficiency of LoRA-FA diminishes with larger models?**
> > > - A.6: When benchmarking the memory consumption in fine-tuning Llama2-13B in the last response, we select a small batch size. Here we report the result when batch size is 8. We set the sequence length to 2048, rank to 64. The result is presented in below.
> > >
> > > |                | Llama2-7B | Llama2-13B |
> > > |----------------|-----------|------------|
> > > | LoRA (bs=8)    | 47.58     | 77.13      |
> > > | LoRA-FA (bs=8) | 38.54     | 63.93      |
> > >
> > > This results shows that when fine-tuning larger models such as Llama2-13B, the memory gap between LoRA and LoRA-FA increases from ~9GB to ~13GB, which indicates that LoRA-FA has a good scaling efficiency in save runtime GPU memory when fine-tuning larger models.
> > >
> > > **CONCLUSION**
> > >
> > > From all these results, we can finally conclude that, LoRA-FA having superior accuracy than LoRA under a limited peak memory constraint.

---

> > > > ### Comment · Reviewer_s6YB · 2023-11-23
> > > >
> > > > Dear Author,
> > > >
> > > > Thank you for your response.
> > > >
> > > > From your experiments, it appears that LoRA-FA has some advantages over LoRA in certain cases. However, I remain unconvinced that LoRA-FA consistently outperforms LoRA. Demonstrating the effectiveness of the proposed methodology is a task for the authors rather than the reviewers.
> > > >
> > > > In appreciation of your comprehensive and timely reply, I have decided to increase the score by one level. Yet, I must maintain a stance of rejection at this time, with the hope that the following feedback will guide improvements in your paper:
> > > >
> > > > 1. The paper does not convincingly demonstrate that LoRA-FA consistently surpasses LoRA.
> > > > 2. There is a lack of overall clarity in the paper.
> > > > 3. The experimental setup is inadequately detailed, affecting the reliability of the findings.

---

### Author Response · Authors · 2023-11-20
**General Response: Summary of the Update**

We really appreciate all reviewers for their positive and valuable comments! We are submitting an updated version of the paper. This revision contains more experimental results, together with detailed experiment settings. We also corrected some results in the paper. We look forward to further feedback and discussions from reviewers.

Multiple reviewers have expressed similar concerns regarding the memory consumption of fine-tuning LLaMA-7B in our paper, which we address below.

We have corrected the results in Table 5 and re-conducted the experiment. Specifically, we fine-tuning Llama2-7B with sequence length = 1024. In the new results, we have several findings:

1. LoRA-FA reduces ~13GB memory compared to LoRA in fine-tuning Llama2-7B when batch size is 16.
2. In practice, when fine-tuning Llama2-7B on various hardware, LoRA-FA has better capability than LoRA. For example, on RTX4090-24G, LoRA can only fine-tuning Llama2-7B when $batch size \leq 2$, while LoRA-FA can enlarge the batch size to 4; on A100-40G, LoRA works with batch size = 4, while LoRA-FA works with batch size = 8.

We also give a runtime memory breakdown analysis of fine-tuning Llama2-7B as follows when batch size is 16, to figure out the memory usage percentage of different modules, such as optimizer state and activation memory.

|                    | LoRA        |                | LoRA-FA     |                |
|:------------------:|:-----------:|:--------------:|:-----------:|:--------------:|
|                    | Memory (GB) | Percentage (%) | Memory (GB) | Percentage (%) |
| Hardware researved | 0.7         | 0.9            | 0.7         | 1.1            |
| Pytorch researved  | 0.5         | 0.6            | 0.5         | 0.8            |
| Model in bf16      | 12.6        | 17             | 12.6        | 20.3           |
| LoRA module        | 0.8         | 1.1            | 0.8         | 1.4            |
| Cache              | 5.1         | 6.7            | 4.8         | 7.7            |
| Optimizer state    | 0.7         | 0.9            | 0.4         | 0.4            |
| Activation         | 55.6        | 73.1           | 42.3        | 68.2           |

This result shows that LoRA-FA can significantly reduce the activation memory usage compared to LoRA when fine-tuning Llama2-7B (from 55.6 GB to 42.3 GB). This advantage can either help fine-tuning on consumer level GPUs, or can achieve higher throughput by enlarging batch size.

---

### Meta-Review · Area_Chair_9MbG · 2023-12-04

**Metareview:**

This paper proposes a simplified version of LoRA, in which only a portion of the LoRA component is finetuned. This simplified LoRA (LoRA-FA) is found to perform on par with regular LoRA across several benchmarks models.

This paper's main strength is that it is simple, and backed up by solid improvements. However, there is a critical weakness in that the upper bound on memory savings is actually not that large (GPU memory associated with LoRA parameters is a small fraction of the total GPU memory, even if you include the extra overhead due to optimizer memory). Moreover, there is no memory saving in the final model, which could (for example) be achieved if portions of the LoRA parameters were shared across layers.

**Justification For Why Not Higher Score:**

This paper is not targeting an actual problem, since memory-savings during finetuning are small compared to regular LoRA.

**Justification For Why Not Lower Score:**

N/A

---

### Decision · Program_Chairs · 2024-01-16

Reject